# Assessing the In Vitro and In Vivo Effect of Supplementation with a Garlic (*Allium sativum*) and Oregano (*Origanum vulgare*) Essential Oil Mixture on Digestibility in West African Sheep

**DOI:** 10.3390/vetsci10120695

**Published:** 2023-12-07

**Authors:** Olga Teresa Barreto-Cruz, Juan Carlos Henao Zambrano, Roman David Castañeda-Serrano, Lina Maria Peñuela Sierra

**Affiliations:** 1Block 5 Laboratory of Animal Nutrition, Veterinary Medicine and Animal Science Program, Department of Animal Production, University Cooperative of Colombia, Ibague 730003, Colombia; juan.henaoz@campusucc.edu.co; 2Department of Animal Production, University of Tolima, Santa Helena 42 Street n 2, Ibague 730006, Colombia; rcastaneda@ut.edu.co (R.D.C.-S.); lmpenuelas@ut.edu.co (L.M.P.S.)

**Keywords:** additives, bioactive compounds, carvacrol, fiber digestion, sulfur compounds, thymol

## Abstract

**Simple Summary:**

In ruminants, fermentation control plays a crucial role in optimizing feed utilization efficiency and reducing methane emissions. Traditional approaches involving antibiotics and feed additives have been used to modify ruminal fermentation, contributing to the emergence of antibiotic resistance in humans. The use of essential oils as natural additives could potentially replace antibiotics and synthetic feed additives, promoting sustainability in livestock production. The objective of the present study was to determine the optimal dosage of a mixture of garlic and oregano essential oils as feed additives in improving ruminal fermentation. The results showed significant improvements in digestibility with the inclusion of essential oils. Garlic and oregano essential oils have the potential to modulate ruminal fermentation, improving productivity while reducing the reliance on antibiotics. These findings highlight the potential of essential oils in optimizing ruminal fermentation and their contribution to the development of sustainable animal production in ruminants.

**Abstract:**

This study assessed the impact of a mixture of garlic (*Allium sativum*) and oregano (*Origanum vulgare*) essential oils (EOGOs) on in vitro dry matter digestibility (IVDMD) and in vivo apparent nutrient digestibility. Different EOGO inclusion levels were evaluated to assess the dose response and potential effects of the mixture. Three EOGO inclusion levels (0.5, 0.75, and 1 mL/kg of incubated dry matter) were evaluated in vitro, while four treatments (0.5, 0.75, and 1 mL/day of EOGO and a control group) were tested in vivo on 12 West African sheep. A randomized controlled trial was conducted using a 4 × 4 design. Blood parameters (glucose, blood urea nitrogen, and β-hydroxybutyrate) were measured to observe the effect of EOGO on the metabolism. The results showed that the inclusion of EOGO significantly enhanced IVDMD at low levels (*p* < 0.052) compared with the highest levels in treatments containing 0.5 and 0.75 mL/kg of EOGO dry matter. A higher intake of dry matter (DM), crude protein (CP), and neutral detergent fiber (NDF) (*p* < 0.05) was observed in the in vivo diets with the inclusion of EOGO. In terms of in vivo apparent digestibility, significant differences were found among treatments in the digestibility coefficients of DM, CP, and NDF. EOGO inclusion increased the digestibility of DM. CP digestibility displayed a cubic effect (*p* < 0.038), with the lowest values of digestibility observed at 1 mL EOGO inclusion. Additionally, NDF digestibility showed a cubic effect (*p* < 0.012), with the highest value obtained at 0.75 mL of EOGO inclusion. The inclusion levels above 0.75 mL EOGO showed a cubic effect, which indicates that higher concentrations of EOGO may not be beneficial for the digestibility of CP and NDF. Although no significant difference was observed in total digestible nutrients, a linear trend was observed (*p* < 0.059). EOGO improved the intake of DM, CP, and NDF. EOGO supplementation improved the digestibility of DM and NDF, with optimal levels observed at 0.5 mL/day. No significant effects were observed in the blood parameters. These results suggest that EOGO has the potential as an additive in ruminal nutrition to improve food digestibility and serve as an alternative to antibiotic additives. The use of EOGO potentially improves fiber digestion and may reduce the use of antibiotics in livestock production. Garlic (*A. sativum*) and oregano (*O. vulgare*) essential oils effectively modulated fiber digestibility at 0.75 mL/day. Garlic (*A. sativum*) and oregano (*O. vulgare*) essential oils have the potential to improve digestibility at low inclusion levels and serve as an alternative to antibiotic additives. The effectiveness of essential oils is greater in a mixture and at lower doses.

## 1. Introduction

Improving digestibility through ruminal fermentation is essential for ruminant production to enhance feed efficiency and minimize the emission of enteric methane and nitrogen excretion, which are major pollutants in livestock production [1,2]. Antibiotics that control ruminal fermentation play a role in reducing methane production and regulating the rate of fermentation of soluble carbohydrates, thereby improving fiber digestion and regulating the rate of protein degradation to preserve amino acids [3,4]. However, there are global concerns regarding the emergence of antibiotic resistance [5,6,7], with over 70% of antibiotics being used worldwide [8,9] as non-therapeutic antimicrobials in animal nutrition, which is considered one of the most important factors affecting the emergence of antibiotic-resistant bacteria in humans [10,11]. Natural additives, such as essential oils (EOs) [12,13], are a collection of secondary compounds [14,15] with antimicrobial and antibiotic properties [16,17] that could potentially modulate ruminal fermentation [18,19] and improve the nutritional properties of meat and milk [20,21,22,23,24].

Garlic (*Allium sativum*) and oregano (*Origanum vulgare*) EOs, or their main components (thymol, carvacrol, sulfur compounds, and allicin), could improve ruminal fermentation by reducing methane production [25,26,27,28], improving fiber digestion [29,30,31], and modulating ruminal populations and fermentation [32,33,34].

The effectiveness of EOs as additives may be greater at lower doses [29,30,35,36,37] and when used as a mixture for their synergistic effects [13,21,33,38,39], whereas higher doses could be detrimental to ruminal fermentation and populations [14,23,40,41]. Most previous studies on the effects of EOs on ruminants were conducted in vitro [42,43,44,45,46,47,48,49].

The active compounds in EOs can vary due to factors such as climate and the plant part used [50,51]. Garlic-oil-derived compounds containing organic sulfur, such as diallyl sulfide, diallyl disulfide, diallyl trisulfide, and allicin, have antimicrobial properties [37,52,53,54].

Carvacrol is the main component of oregano EO [55,56] and has been shown to have antimicrobial [47,55], antioxidant [57,58,59], and anti-inflammatory properties [57,60]. In addition, other compounds present in EOs, such as ρ-cymene and limonene, also have antimicrobial and anti-inflammatory properties [58,61].

Dietary supplementation with EOs could improve energy balance in small ruminants, where serum glucose and β-hydroxybutyrate (BHB) can be used as reliable indicators of ruminants’ energy status, while blood urea nitrogen (BUN) can indicate nitrogen metabolism [32,62,63]. While some studies have suggested that EO supplementation may improve metabolic health and energy metabolism [23,64,65], others have reported no significant effects on blood parameters [66,67,68,69,70]. Thus, it is necessary to determine the effect of EO supplementation on energy utilization, fat mobilization, and overall metabolic processes.

It is necessary to elucidate the effect of garlic and oregano essential oil (EOGO) on the complex interactions within an organism and determine safe and effective doses. Optimal doses and inclusion rates may vary depending on the specific animal species, diet composition, and production goals, and further research is required to fully understand their potential benefits and limitations in ruminant nutrition.

This study aimed to evaluate the impact of a combination of EOGOs on the in vitro dry matter digestibility (IVDMD) and in vivo digestibility of dry matter (DM), crude protein (CP), and neutral detergent fiber (NDF) in West African sheep.

## 2. Materials and Methods

### 2.1. Study Location and Animal Care

This study was conducted in the tropical dry forest conditions of Colombia at an altitude of 1168 m above sea level (4°25′59″ N, 75°13′1″ W). The research protocol and all animal care procedures were approved by the Ethical Committee for Animal Research at the University Cooperative de Colombia (Bioethics Committee Act Number 0316). The animal handling and experimental procedures were performed according to the guidelines for the care and use of animals in research. Special care was taken to ensure that the animals were not subjected to any unnecessary stress or discomfort during the study.

### 2.2. In Vitro Digestibility and Dose Selection

Prior to conducting the in vivo experiment, an in vitro digestibility protocol was performed to determine the appropriate dose of the combination of EOGOs. The protocol followed the guidelines for the DAISYII^®^ incubator (ANKOM Technology, Fairport, NY, USA), using Ankom FN° 57 bags with 0.5 g of sample per bag. For each treatment, 24 bags were conditioned in four glass jars with a volume of 2000 mL, including one blank bag (empty and sealed), to determine the correction factor for the possible entry of particles or the weight loss of the bags. The rumen inoculum was collected with the help of a vacuum bomb and transported with CO_2_ until incubation. This was collected from a cannulated bovine feeder with the same diet proportions to obtain a microbiota ratio similar to that of the experiment (21 days before collection). The bags were incubated for 48 h at 39.2 ± 0.5 °C. Three levels of EOGO inclusion were evaluated and distributed across four treatments: Treatment 1—control (without added additives), Treatment 2—0.5 mL, Treatment 3—0.75 mL, and Treatment 4—1 mL EOGO per kg of DM. These dosages were equivalent to the amount of treatment applied to each jar containing 12.5 g of DM, which was then incubated. The same diet was used for the in vivo and in vitro conditions, and the diet ratio consisted of a 60:40 proportion of forage:concentrate (corn silage and corn, soybean cake and mineral supplement, respectively).

### 2.3. Gas Chromatography–Mass Spectrometry Analysis of Secondary EO Compounds

Sample analysis was performed using gas chromatography coupled to mass spectrometry (GC-MS), with the certified mixture of Cis hydrocarbons (AccuStandard, New Haven, CT, USA) as the reference standard. Sample preparation involved the dilution and direct injection of the EOs into the chromatographic equipment. Chromatographic analysis was performed using an Agilent Technologies AT 6890 Series Plus gas chromatograph (Agilent Technologies, Palo Alto, California, USA), coupled with an Agilent Technologies MSD 5975 mass selective detector operating in full scan mode with radio frequency. The column used for the analysis was a DB-5MS (J&W Scientific, Folsom, CA, USA) with a 5%-phenyl-poly(dimethylsiloxane) phase, measuring 60 m × 0.25 mm × 0.25 µm. The injection was performed in split mode (30:1) with an injection volume of 2 µL.

The EO samples were analyzed for their chemical constituents using mass spectrometry, with electron ionization at 70 eV. The Adams, Wiley, and NIST databases were employed to identify the compounds detected. Table 1 and Table 2 present the retention times, relative quantities (%), and identities of the components identified in the EOs. In the case of the garlic (*A. sativum*) EO, the major constituents were diallyl trisulphide (25%), diallyl disulfide (22.7%), diallyl monosulfide (7.3%), and tetrasulfide of diallyl (6.7%). For the oregano (*O. vulgare*) EO, the principal components were carvacrol (79.4%), thymol (6.9%), and ρ-Cymene (4.0%).

### 2.4. Animals and Diet for In Vivo Experiment

Twelve intact male West African sheep with a mean body weight of 20 ± 2.5 kg (mean ± standard deviation) and approximately three months old were included in the study. They were housed in pens with three animals per pen and fed twice daily with ad libitum access to water. A 4 × 4 Latin square design was used over four periods. The animals were fed the same base diet (Table 3) twice a day: in the morning at 8:30 a.m. and in the afternoon at 4:30 p.m., with an allowance of 5–10% leftovers based on the natural matter of the offered feed. The base diet was formulated according to [71]. The diet consisted of corn silage and concentrate (corn, soybean meal, molasses, and dicalcium phosphate mineral premix) at a 60:40 ratio based on the DM. The EOGOs were orally administered daily to ensure consumption and were mixed in equal parts (*v*/*v*). The mixture was subsequently administered at a volume of 5 mL using glycerol as a vehicle. In the case of the control treatment, 5 mL of glycerol was administered without the inclusion of EOGOs.

### 2.5. Treatments and Experimental Periods

According to the obtained data on in vitro digestibility, the animals were assigned to one of four treatments, i.e., Treatments 1–4. The experimental period lasted for 16 days, where the first 12 days allowed for acclimation to the experimental diets, and the last 4 days were reserved for sample collection in individual metabolic cages. During sample collection, the animals were housed in digestive cages, and the total fecal production was collected on days 13–16 of the experimental period, twice a day at alternate times with a 4 h interval. The leftovers were weighed, homogenized and sampled (10% per animal during each evaluation period) for chemical analysis along with the offered food. The chemical composition was determined using the same methods as for that of DM (number 930.15), CP (number 992.15), ether extract (number 920.39) [72], and NDF [73], with addition of amylase [74]. The total digestible nutrient (TDN) was calculated according to the equation proposed by Sniffen [75]. All samples were pre-dried in an oven at 55 °C for 72 h and subsequently ground to a 1 mm thickness using a mill. The samples were collected four times per animal per treatment per experimental period, which resulted in a total of 16 samples per animal for analysis.

### 2.6. Blood Parameters

After a 12 h fasting period, blood was collected from the jugular veins of the animals by venepuncture. A total of 5 mL of whole blood was collected, with EDTA as the anticoagulant. The samples were refrigerated until they were processed in the laboratory, where they were centrifuged at 1500 rpm for 15 min to obtain the plasma. Enzymatic and colorimetric tests were performed to determine the plasma glucose, BUN, and BHB concentrations [76].

### 2.7. Statistical Analysis

The experimental design was a 4 × 4 Latin square. An analysis of variation (ANOVA) was performed using the mixed model methodology in MINITAB 17™ [77]. For variables that were repeated over time, a split-plot arrangement was used (subdivided plots), considering the effect of time and the interaction between time and treatment. The effects of the EO inclusion levels were analyzed using polynomial regression models. The mathematical model used included the period, treatment, and animal effects: Yijk = μ + Ai + Pj + Tk + eijk, where μ = mean of the treatments; Ai = effect of the i animal, ranging from 1 to 4; Pj = effect of the j period, ranging from 1 to 4; Tk = effect of the k treatment, ranging from 1 to 4; and eijk = random error. The effects of the periods and the interaction between treatments and periods were defined using a Fisher test applied to the ANOVA. Effects were determined to be significant at *p* < 0.05.

## 3. Results

### 3.1. In Vitro Digestibility

The inclusion of a mixture of EOGOs had a positive effect on the IVDMD at low inclusion levels (*p* < 0.034), with values of 64.51c, 73.72a, 71.44b, and 66.36c for the control, 0.5, 0.75, and 1 mL treatments, respectively. Treatments with 0.5 and 0.75 mL presented the highest IVDMD values, while the treatment with 1 mL and the control group presented the lowest values (Figure 1).

A polynomial regression analysis was performed, and it was determined that increasing levels of EOs conferred a quadratic effect (*p* < 0.046 and R-Sq(adj) of 0.986) on the IVDMD (Figure 2). Based on the regression equation, inclusion levels of 0.5, 0.75, and 1 mL/day were selected for the in vivo digestibility study.

Differences were observed between the treatments regarding the intake of DM, CP, and NDF (*p* < 0.05) (Table 4). The treatments with EOGO inclusion showed a higher dry matter intake (DMI) (756 and 912 g DM/day) at the lower doses (0.5 and 0.75 mL) (*p* < 0.05). CP intake was also higher (94.83 and 93.53 g DM/day) in the treatments with the inclusion of 0.5 and 1 mL of EOGO (*p* < 0.05). Differences (*p* < 0.05) were also observed in the consumption of NDF between the treatments, with the consumption of NDF being higher in the diets with an inclusion level of 0.75 mL of EOGO (350.65 g DM/day).

The inclusion of EOGO had an observed quadratic effect on DMI (*p* < 0.033), CP (*p* < 0.002), and NDF (*p* < 0.001) intake, with the highest values occurring at 0.75 mL of EOGO inclusion.

Between treatments, the effect of EOGO was observed in the digestibility coefficients of DM, CP, and NDF (*p* < 0.05) in the EOGO treatment groups (691.4, 737.5, 737.5, and 74.01 g/kg respectively). An effect on the NDF was also observed (*p* < 0.05) (536.8, 560.5, 649.7, and 592.9 g/kg, respectively).

In the EOGO treatment groups, a linear effect (*p* < 0.046) was observed on DM digestibility as the EOGO inclusion levels increased. A cubic effect (*p* < 0.038) was observed on CP digestibility, with the lowest values observed at an inclusion level of 1 mL of EOGO. For NDF digestibility, there was an observed cubic effect (*p* < 0.012), with the highest value occurring in the 0.75 mL treatment group.

### 3.2. Blood Parameters

There were no observed effects of the treatments or inclusion levels (*p* > 0.05) of EOGO on the plasma glucose (mg/dL), BUN (mg/dL) or BHB concentrations (mmol/L) (Table 5).

## 4. Discussion

### 4.1. In Vitro Digestibility

The in vitro testing results indicate that the use of EOGOs at low levels can improve IVDMD. Specifically, at doses of 0.5 and 0.75 mL, the expected DMI values acquired using to the specified equation (y = 64.57 + 33.77x − 32.12 × 2) were accurate. The expected values were 73.43% and 71.83% DMI, and the obtained values were 73.75% and 72.75% DMI for the 0.5 and 0.75 mL treatments, respectively. However, for the control and 1 mL doses, there was a variation of more than 5%, which can be attributed to biological and individual responses. It should be emphasized that in vitro studies do not fully account for the complex interactions that occur within an organism, and therefore, EOs have the potential to enhance ruminal fermentation by positively impacting volatile fatty acid (VFA) concentrations, inhibiting methane (CH4) production and reducing ammonia nitrogen (NH3-N) concentrations [48,57,70,78]. The effects on the ruminal fermentation can vary between studies, exhibiting no significant impact, positive effects, or negative effects [27,48,66,68,78,79,80,81]. These findings highlight the importance of determining the optimal doses depending on the type of diet and investigating the biological effects of the adaptation of ruminal microbiota.

The in vitro testing indicates that the EOGOs can improve IVDMD at low levels. It has been suggested [34] that supplementation with oregano EO can modify the ruminal fermentation to alter the VFA concentrations and reduce methane emissions by altering the ruminal bacterial community at low doses (52 mg/L), thereby improving digestibility.

Garlic EO effectively lowered methane production [43], decreased the abundance of methanogens, and altered the abundances of several bacterial populations that are important for in vitro feed digestion at a concentration of 0.50 g/L. Similar findings [82] regarding evaluations of garlic EO at the lowest level of inclusion (167 µL/L) found this dose to be the most appropriate, as higher doses were detrimental to feed digestibility and fermentation. At this level, the garlic EO exhibited the highest methane inhibition (38.5%). In addition, the inclusion of garlic EO at 167 µL/L resulted in a significant increase in the total VFA and propionate production and a decreased ratio of acetate to propionate but had no effect on feed digestibility. These results suggest that the garlic EO has the potential to mitigate methane production without negatively affecting feed digestibility when used as a feed additive. However, further research is required to determine the optimal dose and evaluate its effects on animal performance and health.

### 4.2. In Vivo Experiment

Our data suggest that the effect of EOGOs on digestibility can be explained by their impact on ruminal bacterial populations. Data from various studies have indicated that the susceptibility of bacteria to EOs primarily resides in the bacteria’s cell wall [21,83,84,85]. Thymol has been shown to induce changes in membrane permeability, which lead to the release of potassium ions (K^+^) and ATP [54,86]. Consequently, changes in the growth rate directly impact the composition and proportion of bacterial populations in the rumen, particularly gram-negative bacteria [87,88,89]. Monensin and EOs could have distinct effects on the composition of the rumen microbiota. In the rumen microbiota of transition dairy cows, it was found that a blend of EOs (thymol, guaiacol, eugenol, vanillin, salicylaldehyde, and limonene) did not significantly affect the microbiota; however, the study demonstrated that monensin sensitivity could be influenced by the structure and thickness of the bacterial cell wall rather than a clear differentiation between gram-negative and -positive bacteria.

The intake of DM, CP, and NDF increased with the inclusion of EOGOs at the 0.75 and 1 mL treatment levels. This increase can be attributed to improved fiber digestibility, which results in a higher rate of ruminal passage and subsequent increased intake [90]. These findings are consistent with previous studies on EOs. For example, a study [91] on cannulated grazing beef cattle using a 4 × 4 Latin square design and a blend of cashew, castor, and copaiba EOs at concentrations of 150, 300, and 450 mg/kg of DM, compared to monensin at 150 mg/kg of DM, showed that at EO lower concentrations (150 mg/kg), NDF digestibility increased and nitrogen utilization efficiency improved. We observed better DM, CP, and NDF digestibility at lower levels of EOGO inclusion. Specifically for DM digestibility, increasing levels of EOGO inclusion led to increased digestibility. For NDF and CP, the maximum digestibility was observed at an inclusion level of 0.75 mL of EOGO per day. These findings demonstrate that EOGOs can effectively modulate fiber digestibility at low levels of inclusion and that there is an effect when used in combination, which allows for the effective modulation of ruminal fermentation at low levels.

It was observed that the inclusion of 1 mL of EOGO resulted in decreased CP digestibility. EOs can inhibit specific ruminal populations, which leads to the inhibition of deamination and subsequently affects CP digestibility [56,92,93]. These findings are consistent with previous studies that have shown the ability of EOs to modulate protein degradation and improve digestibility, particularly at lower doses [12,36,69,94].

The effect on fiber digestibility could be explained by the effect of EOs on ruminal bacterial populations. In fistulated German Merino sheep, it was found that supplementation with oregano EO at a low dose of 4 g/day led to an increase in the populations of *Ruminococcus flavefaciens, Ruminococcus albus*, and *Fibrobacter succinogenes,* which suggests that oregano EO selectively promotes the growth of specific ruminal microbial populations. However, supplying high doses of oregano EO may have a negative impact on the same ruminal microbial populations [34].

In black-brown Swiss mountain sheep and Holstein cows, the inclusion of garlic EO in the diet did not affect NDF digestibility [31,53]. In an in vitro study with a 50:50 forage:concentrate ratio, the inclusion of 300 mg/L of garlic EO did not affect NDF digestibility [95,96]. The addition of EOs from *Anacardium occidentale* and *Ricinus communis* at inclusion rates of 1, 2, 4, and 8 g/day to a high-forage diet (80% *Brachiaria humidicola hay*) resulted in improved fiber digestion and digestibility, among which the greatest improvement was observed at the lowest dose (2 g/day) [29].

It has been suggested that EOs have a greater effect at lower doses and in combinations, while high doses may have a deleterious effect on fiber digestibility due to their antimicrobial properties [49,97].

Other authors have observed that increased levels of EOs can decrease total digestibility. EOs have the potential to decrease the deamination of amino acids through their effect on ammonia-producing bacteria and protozoa [14,65]. The effect on NDF digestibility can be attributed to the control of the rumen bacterial populations.

The presence of organosulfur compounds in garlic EO may have a direct inhibitory effect on rumen methanogenic archaea by inhibiting the enzyme 3-hydroxy-3-methyl-glutaryl coenzyme A reductase [96]. In addition, it has been found that oregano EO decreases ruminal protozoa, indicating that oregano EO could inhibit the protozoa, thereby affecting protein degradation in the rumen [34].

In the present study, the inclusion of low levels of EOGOs (0.5 and 0.75 mL/day) in the diet improved DM and NDF digestibility and decreased the degradation of CP. The inclusion of EOGO positively modified the rumen microbiota by modifying the degradation of CP and fiber.

### 4.3. Blood Parameters

No effects on the blood parameters were found in the present study. For BHB, similar findings have been observed when supplementing the high-concentrate diets of feedlot cows with thyme or cinnamon EOs, which did not significantly affect the blood parameters, including glucose, cholesterol, triglyceride, urea-N, BHB, alanine aminotransferase, and aspartate aminotransferase [66].

However, it was observed in dairy cows fed with 1.2 g of a blend of EOs (containing menthol, eugenol, and anethol) that the BHB levels decreased with EO inclusion [98]. Similar data were reported [68] in dairy cows that were supplemented with a combination of capsicum oleoresin and clove EO, which resulted in a quadratic decrease in serum BHB, indicating improved metabolic health. The serum insulin concentration was also decreased in primiparous but not multiparous cows. However, nutrient utilization and other blood parameters were not affected.

The elevated glucose concentrations (102 mg/dL) that were observed in this study may be attributed to stressors during the days of sample collection, environmental conditions, and genetics. Stress activates the pituitary–adrenal axis, leading to the release of cortisol, which induces a hyperglycemic effect [99,100]. These values are in accordance with the expected results based on the genetic crosses and environmental conditions of our study. In hair sheep under tropical conditions, it was reported that the average glucose level was 98.4 mg/dL [101]. Similarly, an increase in glucose values was observed near parturition (164.90 ± 136.52 mg/dL) in Santa Inés sheep in Brazil [102].

Overall, the effects of EOs on the blood parameters and BHB levels may vary depending on the type and dose of EOs used, as well as the specific conditions of the study. Additional research is required to assess the effects of EOs on the energy balance of ruminants.

## 5. Conclusions

Our in vitro and in vivo experiments suggest that EOGOs have a positive and synergistic effect on digestibility in ruminants at low doses. EOGO inclusion levels of 0.5 and 0.75 mL per animal per day led to higher DM and NDF digestibility, with the maximum digestibility observed at 0.75 mL for NDF. These findings suggest that EOGOs have the potential to improve ruminal fermentation and fiber digestibility in ruminants, which could have important implications for ruminant production. Specifically, the use of EOGOs at low inclusion levels could lead to increased feed efficiency and animal performance. Further research is required to fully understand the potential benefits and limitations of the use of EOs. EOGOs enhance fiber digestion in ruminants and can be used as ruminal additives, potentially replacing antibiotics in ruminal nutrition.

Further studies are necessary to better understand the potential impact of EO supplementation on the various aspects of ruminant metabolism. This includes investigating how EO supplementation may influence the microbiota, energy utilization, fat mobilization, and overall metabolic processes.

## Figures and Tables

**Figure 1 vetsci-10-00695-f001:**
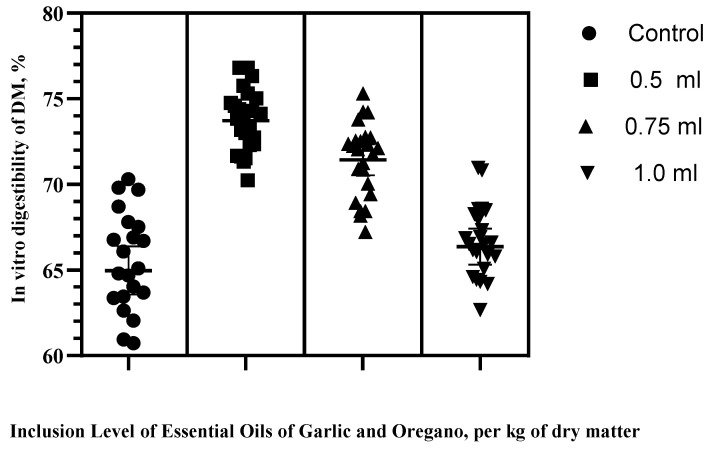
In vitro digestibility of dry matter (IVDM) with different inclusion levels of a mixture of garlic (*Allium sativum*) and oregano (*Origanum vulgare*) essential oils.

**Figure 2 vetsci-10-00695-f002:**
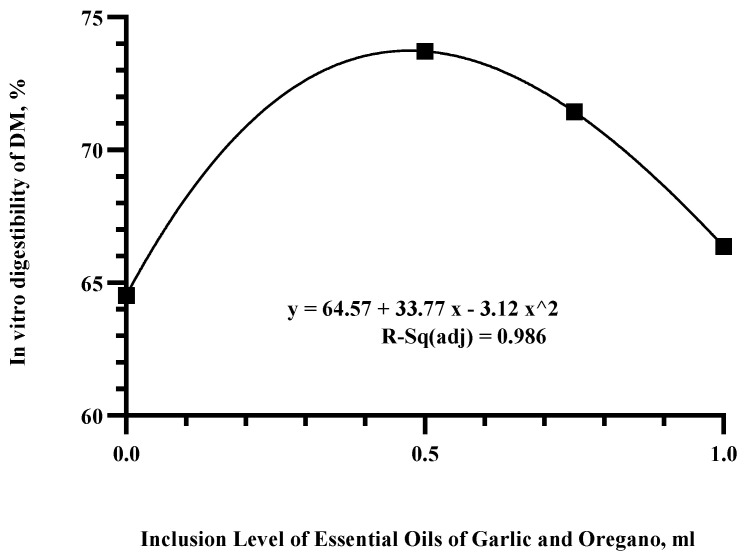
In vitro digestibility of dry matter (IVDM) polynomial analysis.

**Table 1 vetsci-10-00695-t001:** Identification, retention times, and relative amount (%) of the secondary compounds present in the garlic (*Allium sativum*) essential oil, identified via GC-MS.

Retention Time, min (t_R_)	Compound Identification	Relative Amount, %
7.56	Allyl methyl sulfide	0.50
9.03	dimethyl disulfide	0.20
13.75	diallyl monosulfide	7.30
16.3S	sharp methyl disulfide	3.30
18.79	dimethyl trisulfide	0.60
23.57	diallyl disulfide	22.70
24.08	cis-propenyl-propyl disulfide	0.10
25.99	Methyl allyl trisulfide	9.70
26.8	4-Methyl-1,2,3-trithiolane	2.70
29.05	dimethyl tetrasulfide	0.70
29.29	* Compound NI m/z (%): 162 (4), 121 (23), 89 (55), 75 (100), 59 (12), 41(88)	0.70
31.49	3-Ethyl-2,4,5-trithiahexane	0.70
32.21	diallyl trisulfide	25.00
35.16	5-Methyl-1,2,3,4-tetrathian	2.00
35.36	* Compound NI m/z (%): 184 (10), 158 (15), 143 (1), 120 (34), 94 (4), 79 (45), 64 (41), 41 (100)	0.80
36.9	1-Methyl-2-(1-(prop-1-en-1-ylthio)propyl)disulfane	0.30
37.49	1-(1-{Methylthio)propyl)-2-propyl-disulfane	2.40
39.57	4-Ethyl-2,3,5,6-tetrathiaheptane	0.30
40.75	diallyl tetrasulfide	6.70
41.85	1-Methyl-2-(2-propenylthio)ethyl-2-propenyl disulfide	1.00
42.13	1-propenyl 1-(1-propenylthio)propyl disulfide	2.70
44.01	* Compound NI m/z(o/o): 202 (11), 170 (28), 138 (52), 106 (17), 96 (15), 64 (71), 41(100)	1.90
44.4	* Compound NI m/z(o/o): 202 (1), 170 (13), 138 (9), 121 (69), 106 (7), 89 (34), 73 (67), 41 (100)	0.90
46.5	* Compound NI m/z(o/o): 192 (1), 177 (3), 145 (4), 113 (100), 99 (14), 85 (36), 79 (68), 64 (21), 41 (72)	2.20
47.11	1,5-Dithiaspiro[5.6]dodecan-7-ol	0.90
47.84	8-Methyl-4,5,6, 9-tetrathia-1, 11-dodecadiene	3.70

* Presumptive identification.

**Table 2 vetsci-10-00695-t002:** Identification, retention times, and relative amount (%) of the secondary compounds present in the oregano (*Origanum vulgare*) essential oil, identified via GC-MS.

Retention Time, min (t_R_)	Compound Identification	Relative Amount, %
19.52	β-Myrcene	0.30
20.26	ρ-Mint-1(7),8-diene	<0.1
20.81	α-Terpinene	0.20
21.17	ρ-Cymene	4.00
21.36	Limonene	1.00
22.55	γ-Terpinene	0.60
24.2	Linalool	1.60
25.41	(1R,2S,3S)-3-Isopropenyl-1,2-dimethylcyclopentanol	<0.1
31.45	Thymol	6.90
32.11	Carvacrol	79.40
35.04	α-Copaene	0.10
36.75	trans-β-Caryophyllene	2.10
37.96	α-Humulene	0.20
39.82	δ-Cadinene	<0.1
41.95	caryophyllene oxide	1.90
42.38	Humulene epoxide I	<0.1
42.71	humulene epoxide II	0.10
44.22	(1R,7S,E)-7-isopropyl-4,10-dimethylene-cyclodec-5-enol	0.20
50.39	5-(6-Methylhepta-1,5-dien-2yl)1-1-(4-methylpent-3-en-1-yl)cyclohex-1-ene (m-Camphorene)	0.10
51.14	4-(6-Methylhepta-1,5-dien-2-yl)-1-(4methylpent-3-en-1-yl)cyclohex-1-ene (ρ-Camphorene)	<0.1
51.89	* Compound NI m/z (%): 150 (53), 135 (100), 121 (13), 107 (14), 93 (30), 79 (16), 65 (8)	0.30
52.41	* Compound NI m/z (%): 150 (72), 135 (100), 121 (12), 107 (11), 93 (23), 79 (15), 65 (8)	0.10
53.8	4a,6a-Dimethyl-4,4a,6,6a,8,9,9a,9b,10,11-decahydrocyclopenta[7,8] phenanthro[4β,5-β]oxirene-2,7(3H,5ah)-diona	<0.1
54.01	* Compound NI m/z (%): 302 (60), 284 (6), 259 (34), 241 (84), 201 (100), 173 (20), 159 (71), 145 (14), 131 (9), 115 (17), 91 (18), 58 (20)	0.20
54.49	(Z)-2Methyl-6-(4-methyl-5-(3-methylbut-2-enoyl)cyclohex-3-en-1-yl)hepta-2,5-dien-4-one	0.20
54.66	Androsta-1,4,7-triene-3,17-dione	<0.1
63.25	* Compound NI m/z (%): 370 (18), 355 (1), 221 (5), 203 (32), 175 (8), 150 (100), 135 (92), 121 (21), 107 (29), 93 (25), 79 (28)	0.30

* Presumptive identification.

**Table 3 vetsci-10-00695-t003:** Chemical composition of the total mix ration (TMR) and percentage composition of ingredients used in the experimental basal diet.

Ingredients and Chemical Composition of the Diets	
	% of DM
**Ingredient**	
Maize silage	60.00
Corn grain, ground	21.00
Soybean meal	17.00
Molasses	1.10
Bicalcium phosphate	0.01
Mineral mixture ^1^	0.89
**Chemical composition**	
Crude protein (CP)	11.55
Neutral detergent fiber (NDF)	40.63
Ether extract (EE)	3.45
Total digestible nutrients (TDN)	68.00

^1^ Calcium: 130.0 g (max.), phosphorus: 65.0 g (min.), sodium: 135.0 g, sulfur: 12.0 g, magnesium: 12 g, manganese: 1.050 mg, cobalt: 63 mg, iodine: 63 mg, copper: 1.155 mg, selenium: 18 mg, zinc: 3.080 mg, eFluor: 650 mg. Vitamin premix.

**Table 4 vetsci-10-00695-t004:** In vivo total apparent consumption and digestibility in West African sheep supplemented with different levels of garlic (*Allium sativum*) and oregano (*Origanum vulgare*) essential oils.

	^+^ EO Inclusion mL/day					*p*-Value		
Item	0	0.5	0.75	1.0	SEM ^1^	L ^2^	Q ^3^	C ^4^
Dry matter	587.79 c	756.27 b	912.87 a	889.94 a	14.06	0.070	0.033	0.394
Crude Protein	74.60 c	94.83 b	107.11 a	93.53 b	2.20	0.063	0.002	0.427
Non-Fiber Carbohydrates	332.73	315.54	331.84	320.37	6.29	0.825	0.891	0.514
Neutral Detergent Fiber	217.00 c	306.44 b	350.65 a	325.16 ab	12.74	0.062	0.001	0.738
Ether Extract	30.05 ab	29.21 b	29.86 ab	30.77 a	0.53	0.743	0.650	0.887
Dry matter	691.4 b	737.5 a	727.5 a	731.4 a	0.97	0.046	0.085	0.200
Crude Protein	671.7	629.6	666.4	616.8	1.66	0.105	0.831	0.038
Non-Fiber Carbohydrates	901.3	906.0	887.5	896.0	1.24	0.534	0.897	0.365
Neutral detergent fiber	536.8 b	560.5 b	649.7 a	592.9 b	2.15	0.076	0.062	0.012
Ether extract	787.1 bc	821.1 ab	785.1 c	823.6 a	1.18	0.473	0.922	0.162
TDN ^5^	698.5	707.3	737.5	702.9	1.10	0.391	0.059	0.092

^1^ SEM: standard error of means. ^2^ L = linear; ^3^ Q = quadratic; ^4^ C = cubic; ^5^ TDN = total digestible nutrients. Means with different letters show statistical differences according to the Fisher test. Letters (a, b, c, ab, bc) denote treatment distinctions. Identical letters signify nonsignificant differences, while differing letters denote statistical significance. ^+^ Garlic essential oil comprising 25% diallyl disulfide and 22% diallyl monosulfide, and oregano essential oil comprising 79% carvacrol and 6.9% thymol.

**Table 5 vetsci-10-00695-t005:** Plasma glucose (mg/dL), blood urea nitrogen (mg/dL), and β-hydroxybutyrate concentration (mmol/L), of West African sheep supplemented with different levels of garlic (*Allium sativum*) and oregano (*Origanum vulgare*) essential oils.

	^+^ EO Inclusions mL/day					*p*-Value		
Item	0	0.5	0.75	1.0	SEM ^1^	L ^2^	Q ^3^	C ^4^
Glucose, mg/dL	98.92	101.67	105.67	102.83	2.611	0.227	0.336	0.532
BUN, mg/dL	20.78	18.68	18.78	19.06	1.026	0.298	0.276	0.674
β-hydroxybutyrate, mmol/L	0.42	0.37	0.37	0.37	0.036	0.523	0.634	0.831

^1^ SEM: standard error of means. ^2^ L = linear; ^3^ Q = quadratic; ^4^ C = cubic. ^+^ Garlic essential oil comprising 25% diallyl disulfide and 22% diallyl monosulfide, and oregano essential oil comprising 79% carvacrol and 6.9% thymol.

## Data Availability

The original data presented in the study are included in the present article; further inquiries can be directed to the corresponding author.

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
