# Peer review of "Assessing the In Vitro and In Vivo Effect of Supplementation with a Garlic (Allium sativum) and Oregano (Origanum vulgare) Essential Oil Mixture on Digestibility in West African Sheep"

_vetsci, 2023, doi:10.3390/vetsci10120695_

Round 1
Reviewer 1 Report
Comments and Suggestions for Authors
Title: Title should be extended based on the content of paper.
Explain why you mixed oregano oil and garlic oil, why not other essential oils in the introduction section.
L32-36: These lines are not clear to me why they are presented.
L73: what is EOGO? Define when used for the first time.
L143: does it mean that it was not mixed with feed, but it was directly put in the mouth?
L170: what is jejune?
L191 and elsewhere: use maximum three decimal places for the p-values.
L191-193: donot use comma for the decimal places.
Table 4 and 5: use period (.) for the decimal places. Present actual values instead of NS.
Table 5: Glucose level is too high for ruminants. It is usually within 60 mg/dl. Check the results and calculations.
L277: please discuss on the effect of EOGO on feed intake. What was the reason of increased feed intake.
L297-299: "In a study conducted by Jiao in 2021 using a 3x3 Latin square 297
design, feeding a combination of cobalt lactate and an essential oil blend containing oregano at 4 or 7 g/day resulted in an increase in NDF digestibility in cannulated Merino sheep fed a forage: concentrate diet ratio of 70:30." This is not a suitable context in this study because cobalt was present. Delete it.
L321: Discuss more with some citations.
Please discuss the effect of EOGO on protein digestibility. Why was protein digestibility decreased?
English revision is necessary.
Comments on the Quality of English Language
English revision is necessary.
Author Response
|
Thank you very much for taking the time to review this manuscript. Please find the detailed responses in the attached file. The manuscript was sent for English review, but unfortunately, for this version that you are reviewing, we have not yet received the final revised English version. We will make sure to send the corrected English version to editors as soon as it arrives to us. We have provided this version to meet the 10-day deadline.
The colored highlighted parts in the text correspond to the modifications made in the manuscript.
|
||
|
Point-by-point response to Comments and Suggestions
|
||
|
Comments 1: Title should be extended based on the content of paper. |
||
|
Response 1: Thank you for pointing this out. We agree with this comment. [Line 2. Title “Assessing in vivo and in vitro the effect of Garlic (Allium sativum) and Oregano (Origanum vulgare) essential oil mixture supplementation on apparent digestibility in Vitro and in In vivo West African Sheep” ] We have Extended the point that was in vivo and in vitro [Updated in manuscript in Line 2.]
|
||
|
Comments 2: L32-36: These lines are not clear to me why they are presented. |
||
|
Response 2: We have emphasized the point that in our study at the low levels, it was observed that the EO could improve fiber digestion (0,75ml) and DM digestibility (0,5 ml), which is the major goal to control ruminal fermentation, and the objective to use additives to control ruminal fermentation.
|
||
|
Comments 3: L73: what is EOGO? Define when used for the first time. |
||
|
Response 3: In Line 24 it is defined as a mixture of garlic (Allium sativum) and oregano (Origanum vulgare) essential oils (EOGO). [Manuscript in Line 24]
|
||
|
Comments 4: L143: does it mean that it was not mixed with feed, but it was directly put in the mouth? |
||
|
Response 4: Thank you for pointing this out. Yes, the EOGO was orally administered daily to ensure consumption. Administered in all cases until a volume of 5 ml using glycerol as a vehicle. In the case of the control treatment, 5 ml of glycerol was administered without the inclusion of EOs.
|
||
|
Comments 5: L170: what is jejune? |
||
|
Response 5: Thank you for pointing this out. Revised and Changed. [Line 193. After a 12-hour fasting period] [updated in the manuscript Line 193]
|
||
|
Comments 6: L191 and elsewhere: use maximum three decimal places for the p-values. L191-193: do not use comma for the decimal places |
||
|
Response 6: Revised and Changed. [All document] [updated in all the manuscript]
|
||
|
Comments 7: Table 4 and 5: use period (.) for the decimal places. Present actual values instead of NS |
||
|
Response 7: Revised and Changed in all tables period (.) for decimal places. [All document]
|
||
|
Comments 8: Table 5: Glucose level is too high for ruminants. It is usually within 60 mg/dl. Check the results and calculations. |
||
|
Response 8: Thank you for pointing this out. We have revised and discussed the results in the manuscript. [Line 373 “The elevated glucose concentrations (102 mg/dl) observed in this study may be at-tributed to stressors during the days of sample collection, environmental conditions, and genetics. Stress activates the pituitary-adrenal axis, leading to the release of corti-sol, which exerts a hyperglycemic effect [87,88]. These values are in accordance with the expected results based on the genetic crosses and environmental conditions of our study. In hair sheep under extensive grazing systems in Colombia was reported an av-erage glucose level of 98.4 mg/dL [89]. Similarly, it was observed an increase in glucose values near parturition (164.90 ± 136.52 mg/dL) in Santa Inés sheep in Brazil [90]. Ad-ditionally, Doria et al. (2014) found comparable glucose values (102.42 mg/dl) in crossbred male sheep grazing under tropical conditions in Colombia.”
[Updated in manuscript in Line 373]
|
||
|
Comments 9: L277: please discuss on the effect of EOGO on feed intake. What was the reason of increased feed intake. |
||
|
Response 9: Thank you for pointing this out. We agree with this comment. [Line 298: The intake of DM, CP, and NDF increased with the inclusion of EOGO at the levels of 0.75 and 1.0 ml. This increase can be attributed to improved fiber digestibility, resulting in a higher rate of ruminal passage and subsequently increasing intake [79]. These findings are consistent with other studies on essential oils (EO). For example, a study [80] on grazing beef cattle cannulated in a 4 × 4 Latin-square design, using a blend of cashew, castor, and copaiba essential oils at concentrations of 150, 300, and 450 mg/kg of DM, compared to 150 mg/kg DM of monensin, showed that at the lower concentration (EO 150), NDF digestibility increased, and nitrogen utilization efficiency improved. Similar findings were observed in a study [29] with four Holstein steers fit-ted with ruminal cannula in a 4x4 Latin square design, feeding tropical diets with a high proportion of forage and received EO of cashew and castor at doses of 1, 2, 4 and 8 g/day; the study found that at a dosage of 2 EO g/day, fiber and total digestible nutrients (TDN) digestibility increased]
[Updated in manuscript in Line 298]
|
||
|
Comments 10: L297-299: "In a study conducted by Jiao in 2021 using a 3x3 Latin square design, feeding a combination of cobalt lactate and an essential oil blend containing oregano at 4 or 7 g/day resulted in an increase in NDF digestibility in cannulated Merino sheep fed a forage: concentrate diet ratio of 70:30." This is not a suitable context in this study because cobalt was present. Delete it. |
||
|
Response 10: Thank you for pointing this out. We agree with this comment.
[Updated in manuscript ]
|
||
|
Comments 11: Discuss more with some citations. |
||
|
Response 11: Thank you for bringing this to our attention. We appreciate your comment. We have already expanded the discussion and highlighted it in the manuscript. |
||
|
Comments 12: Please discuss the effect of EOGO on protein digestibility. Why was protein digestibility decreased? |
||
|
Response 12: Thank you for pointing this out. We agree with this comment. [Line 325“It was observed that including EOGO at 1.0 ml resulted in a decrease in CP digestibility. The EO could inhibit specific ruminal populations, leading to the inhibition of deamination and subsequently affecting CP digestibility [56,81,82]. These findings are consistent with previous studies that have shown EO ability to modulate protein degradation and improve digestibility, particularly at lower doses [12,36,83,84].”]
[Updated in manuscript Line 325]
|
||
|
Comments 13: English revision is necessary |
||
|
Response 13: |
||
|
Thank you for your comment. We appreciate your understanding. The manuscript was sent for English review, but unfortunately, for this version that you are reviewing, we have not yet received the final revised English version. We will make sure to send the corrected English version to editors as soon as it arrives to us. We have provided this version to meet the 10-day deadline.
|
||
|
Additional clarifications |
||
|
[For journal editor: We will make sure to send the corrected English version to editors as soon as the English reviewers send the corrected version. We have provided this version to meet the 10-day deadline. |

Reviewer 2 Report
Comments and Suggestions for Authors
The paper "Assessing the effect of Garlic (Allium sativum) and Oregano (Origanum vulgare) essential oil mixture supplementation on apparent digestibility in West African Sheep" deals with an interesting and current topic. The research aims to assess the impact of a mixture of garlic and oregano essential oils on in vitro dry matter digestibility and apparent nutrient digestibility in vivo in West African sheep. The work is quite simple and linear in structure and limited analyses, although appropriate, have been conducted.
The paper needs language revision.
Tables and elesewhere: replace , with .
Line 236-27. Delete measurement, it is not necessary here.
Line 242-43: delete Means with different letters have statistical differences by Fisher test.
There are many inaccuracies and errors e.d. line 253, line 256…) in the text that need to be fixed and corrected.
Line 190: IVDMD was already used before, no need to explain the acronym everytime.
Lines 297 and 300 (and elsewhere): please follow the author’s guideline for citations
Lines 328-330: references are needed here.
Line 341: delete “, including their effects on blood parameters and BHB levels”
Results and discussion should be condensed.
Comments on the Quality of English LanguageThe paper needs English language revision
Author Response
|
Dear Reviewer, Thank you very much for taking the time to review this manuscript. Please find the detailed changes in the attached file. The manuscript was sent for English review, but unfortunately, for this version that you are reviewing, we have not yet received the final revised English version. We will make sure to send the corrected English version to editors as soon as it arrives to us. We have provided this version to meet the 10-day deadline. The colored highlighted parts in the text correspond to the modifications made in the manuscript. |
|
Point-by-point response to Comments and Suggestions
|
|
Comments 1: The paper needs language revision. |
|
Response 1: Thank you for your comment. We appreciate your understanding. The manuscript was sent for English review, but unfortunately, for this version that you are reviewing, we have not yet received the final revised English version. We will make sure to send the corrected English version to editors as soon as it arrives to us. We have provided this version to meet the 10-day deadline.
|
|
Comments 2: Tables and elsewhere: replace , with . |
|
Response 2: Revised and Changed in all tables period (.) for decimal places. [All document]
|
|
Comments 3: Line 236-27. Delete measurement, it is not necessary here. |
|
Response 3: Revised and Changed
|
|
Comments 4: There are many inaccuracies and errors e.d. line 253, line 256…) in the text that need to be fixed and corrected. |
|
Response 4: Thank you for pointing this out. We agree with this comment. [Line 272: “EO (Essential Oil) has the potential to enhance ruminal fermentation by positively im-pact volatile fatty acid (VFA) concentrations, inhibiting methane (CH4) production, and reducing ammonia nitrogen (NH3-N) concentrations [48,57,71,72]. The effects on ruminal fermentation could be changed between studies, including no significant im-pact, positive effects, or negative effects [27,48,72–77]. These findings highlight the importance of determining optimal dosages depending on the type of diet and inte-grating the biological effect of the adaptation of ruminal microbiota.” [updated in the manuscript line 272]
|
|
Comments 5: Lines 297 and 300 (and elsewhere): please follow the author’s guideline for citations |
|
Response 5: Revised and Changed [updated in the manuscript line 343]
|
|
Comments 6: Lines 328-330: references are needed here. |
|
Response 6: Thank you for pointing this out. We agree with this comment. [Line 361: “No effects on blood parameters were found in the present study (P>0,05). For BHB, a similar found was observed supplementing thyme or cinnamon essential oils to a high-concentrate diet in feedlot calves, which did not significantly affect blood pa-rameters, including glucose, cholesterol, triglyceride, urea-N, BHB, alanine ami-notransferase, and aspartate aminotransferase [66]. However, it was observed in dairy cows feeding with 1.2 gr of a blend of essential oils (containing menthol, eugenol and anethol) that BHB levels were decrease with the inclusion of EO [100]. Similar data were reported [68] supplementing dairy cows with a combination of Capsicum oleoresin and clove essential oil resulted in a quadratic decrease of serum BHB, indicating improved metabolic health; serum insulin concen-tration was also decreased in primiparous, but not multiparous, cows. However, nu-trient utilization and other blood parameters were not affected. The elevated glucose concentrations (102 mg/dl) observed in this study may be at-tributed to stressors during the days of sample collection, environmental conditions, and genetics. Stress activates the pituitary-adrenal axis, leading to the release of corti-sol, which exerts a hyperglycemic effect [101,102]. These values are in accordance with the expected results based on the genetic crosses and environmental conditions of our study. In hair sheep in tropical conditions was reported an average glucose level of 98.4 mg/dL [103]. Similarly, it was observed an increase in glucose values near parturi-tion (164.90 ± 136.52 mg/dL) in Santa Inés sheep in Brazil [104]. Additionally, Doria et al. (2014) found comparable glucose values (102.42 mg/dl) in crossbred male sheep under tropical conditions.”] [updated in the manuscript line 361]
|
|
Comments 7: Line 341: delete “, including their effects on blood parameters and BHB levels” |
|
Response 7: Revised and Changed. [updated in the manuscript] Comments 8: Results and discussion should be condensed. |
|
Response 8: Thank you for your feedback. We agree that it is important to condense the results and discussion section to focus on the most relevant and significant information that allows for a clear explanation of our data. We will ensure that as authors, we emphasize only the most important findings in the article. Additionally, we will consider incorporating additional content in the discussion to provide a comprehensive analysis and interpretation of our results.
|
|
Comments 9: English revision is necessary |
|
Response 9: |
|
Thank you for your comment. We appreciate your understanding. The manuscript was sent for English review, but unfortunately, for this version that you are reviewing, we have not yet received the final revised English version. We will make sure to send the corrected English version to editors as soon as it arrives to us. We have provided this version to meet the 10-day deadline.
|
|
Additional clarifications |
|
For journal editor: We will make sure to send the corrected English version to editors as soon as the English reviewers send the corrected version. We have provided this version to meet the 10-day deadline. |

Reviewer 3 Report
Comments and Suggestions for Authors
See attached PDF

Author Response
Dear Reviewer,
Thank you very much for taking the time to review this manuscript. Please find the detailed changes in the attached file. The manuscript was sent for English review, but unfortunately, for this version that you are reviewing, we have not yet received the final revised English version. We will make sure to send the corrected English version to editors as soon as it arrives to us. We have provided this version to meet the 10-day deadline.
The colored highlighted parts in the text correspond to the modifications made in the manuscript.
|
Response to comments.
The manuscript was sent to be proofread to correct grammatical errors.
Abstract We have addressed the revisions in the abstract as follows: Uniformity in unit representation throughout the abstract was corrected. We have briefly mentioned the specific parameters or nutrients (dry matter, crude protein, and neutral detergent fiber) We have provided a brief explanation of the rationale behind selecting the three inclusion levels of EOGO. We have mentioned that the study followed a randomized control trial design (explained that the twelve West African sheep were allocated to treatments using a 4x4 study design) Instead of simply stating "P<0.05," we have specified the actual p-values. We have mentioned the direction of the cubic effect observed for CP and NDF digestibility to provide a more complete picture of the results. We have expanded on the potential benefits of EOGO as an additive in ruminal nutrition. We have briefly discussed the practical implications for livestock producers.
Materials and Methods: We mentioned the mean body weight of the animals and provided additional information on the animal's age.
Results: We discuss the biological significance of the observed changes in digestibility and nutrient intake.
We expanded on the Blood Parameters interpretation of these results. |
|
|
|
Additional clarifications |
|
For journal editor: We will make sure to send the corrected English version to editors as soon as the English reviewers send the corrected version. We have provided this version to meet the 10-day deadline. |
